# Early-Onset Insomnia among Patients with Hemifacial Spasm in South Korea: A Nationwide Cohort Study

**DOI:** 10.3390/jpm13020197

**Published:** 2023-01-22

**Authors:** Young-Goo Kim, Min-Ho Kim, Ga-Eun Kim, Dosang Cho

**Affiliations:** 1Department of Neurosurgery, Ewha Womans University Mokdong Hospital, Seoul 07985, Republic of Korea; 2Informatization Department, Ewha Womans University Seoul Hospital, Seoul 07804, Republic of Korea; 3Department of Psychiatry, Ewha Womans University Mokdong Hospital, Seoul 07985, Republic of Korea; 4Department of Neurosurgery, Ewha Womans University Seoul Hospital, Seoul 07804, Republic of Korea

**Keywords:** hemifacial spasm, psychiatric comorbidity, insomnia

## Abstract

This study aimed to investigate mental illnesses among patients with hemifacial spasms (HFS) based on nationwide claims data from the South Korea Health Insurance Review and Assessment Service. In this retrospective study, we defined the HFS group as subjects aged between 20 and 79 years with newly diagnosed HFS between January 2011 and December 2019 and set the date of diagnosis of HFS as the index date. Mental illnesses were defined through the International Classification of Diseases, the tenth revision from 90 days before to after the index date. Of these patients, we enrolled the participants who had visited a psychiatric outpatient clinic more than twice or had been admitted to a psychiatric department more than once diagnosed with psychiatric diseases. To select the control group, which was four times larger than the HFS group, propensity scores were used among those not diagnosed with HFS. The patients with HFS were more likely to have a mental illness than the control group (8.5% and 6.5%, respectively, *p* < 0.001) within 90 days before and after diagnosis. Among mental illnesses, insomnia (46.2% vs. 13.0%, *p* < 0.001) was significantly more prevalent in the HFS group. Other mental illnesses were significantly more prevalent in the control group or were not statistically significant. The results of this study suggest that patients diagnosed with HFS were significantly more likely to develop insomnia within a relatively short period than the controls.

## 1. Introduction

Hemifacial spasm (HFS) is a motor disorder characterized by unilateral facial muscle contraction. The contractions are intermittent, involuntary tonic, and clonic [1,2]. The symptoms usually start in the orbicularis oculi muscle and are often aggravated in severity and frequency, spreading downward to the ipsilateral facial muscles [3]. The most common cause of HFS is the facial nerve’s compression in its root exit zone by an arterial or venous loop.

Although HFS is not a life-threatening disease, patients with chronic facial twitching may experience social embarrassment and significant suffering in social interaction [4], which may lead to psychiatric comorbidities, including depression [5], anxiety [6], and social phobia [7,8].

However, these results are difficult to generalize because studies investigating HFS have small population samples or cross-sectional designs. Hence, we designed a nationwide, population-based, longitudinal study. This study proposed to investigate the nationwide data on mental illness among patients with HFS within 90 days before and after diagnosis, using claims data from the Health Insurance Review and Assessment Service (HIRA) in South Korea.

## 2. Materials and Methods

### 2.1. Data Sources and Ethics

This study assembled information from the National Health Insurance Service (NHIS) insurance claims database (No. NHIS-2021-1-608) from January 2010 to December 2020. The study was approved by the institutional review board (IRB) of Ewha Womans University (IRB number 2021-02-002). Because this study was a retrospective observational study with an anonymized dataset, informed consent was exempted. The study was carried out according to the Declaration of Helsinki.

The NHIS is compulsory social insurance, covering 97% of the Korean population, whereas the Medical Aid Program covers the remaining 3% who cannot afford national insurance. The HIRA is a quasi-governmental organization established to review claims data and assess the quality of medical care in Korea. Furthermore, the HIRA maintains an electronic database that provides inpatient and outpatient visits in all Korean healthcare institutions, healthcare billing, and reimbursement claims submitted to the Korean National Health Insurance and Medical Aid Program. The HIRA database contains comprehensive epidemiological information on demographics, diagnoses, and medical services. The diagnosis codes are standardized according to the Korean Standard Classification of Disease Version 7 (KCD7), a modified version of the International Classification of Diseases, the tenth revision (ICD-10). Specific information about medications, devices, and medical services was also identified with the help of codes from the HIRA claim database.

As all Korean citizens have a resident registration number from birth to death, exact population statistics can be obtained from the above database. Except in an extraordinary state, all Koreans are enrolled in the NHIS, and all Korean hospitals and clinics register individual patients in the medical insurance system using the resident registration number. For these reasons, the possibility of patient medical record overlap in Korea is extremely low, even if a patient is transferred to another hospital for further treatment. Moreover, the HIRA system can track all medical treatments in Korea. Furthermore, Korean law requires that all deaths be reported to an administrative entity before funerals can be held and causes of death entered on death certificates.

### 2.2. Study Population and Design

In this retrospective study, we defined the HFS group as subjects aged between 20 and 79 years with newly diagnosed HFS (ICD-10: G513) between January 2011 and December 2019 and set the date of diagnosis of HFS as the index date. Participants who died within one year of the index date were excluded. The mental illnesses were defined using the ICD-10 codes (Table 1). To clarify the correlation between HFS and mental illness, we enrolled participants who had visited a psychiatric outpatient clinic more than twice or had been admitted to a psychiatric department more than once with a diagnosis of psychiatric disease for 90 days before and after the index date. Participants diagnosed with any mental illness before 90 days from the index date were excluded.

In addition, for information on patients who have never been diagnosed with HFS, we requested information on patients who were not diagnosed with HFS from January 2002 to December 2020. Among them, 20 times as many patients as the enrolled patients with HFS were selected as non-HFS participants. To select the control group, which was four times larger than the HFS group, propensity scores corresponding to age, sex, socioeconomic status, location of residence, and past medical histories (dyslipidemia, diabetes mellitus, hypertension, and chronic renal disease) were used among those not diagnosed with HFS (Figure 1). The Charlson comorbidity index scores based on the KCD from January 2011 and December 2019 were evaluated. The control and matched HFS participants were assumed to have the same index date.

### 2.3. Variables

The following age groups were defined using 10-year intervals: 20–29, 20–39, 40–49, 50–59, 60–69, and 70–79. A total of six age groups were designated. The socioeconomic status was divided into six classes (covered by medical aid [lowest income] ~ fifth quintile [highest income]). The location of residence was classified into urban (Seoul, Busan, Incheon, Daejeon, Daegu, Gwangju, and Ulsan) and rural (Gyeonggi-do, Gangwon-do, Chungcheongbuk-do, Chungcheongnam-do, Jeollabuk-do, Jeollanam-do, Gyeongsangbuk-do, Gyeongsangnam-do, and Jeju) areas. The past medical histories of participants were assessed using ICD–10 codes. History of hypertension, diabetes, dyslipidemia, and chronic renal disease was defined by at least two claims within one year using the appropriate ICD–10 code.

### 2.4. Statistical Analyses

Baseline patient characteristics and comorbidities are stated as means ± standard deviation for continuous variables and frequency (percentage, %) for categorical variables. The characteristics of study participants were compared using Student’s *t*-tests for continuous variables and the chi-square test for categorical variables. To identify the association between mental illness and the presence of HFS, logistic regression analysis was used to estimate the odds ratio (OR), 95% confidence interval, and *p*-value adjusted for age, sex, socioeconomic status, location of residence, and comorbidities, as well as the Charlson Comorbidity Index (CCI). Additionally, to reduce the impact of potential confounding variables, we conducted propensity-score matching analyses. The propensity scores were calculated non-parametrically using the patient’s age, comorbidities, location of residence, and the Charlson comorbidity index. The propensity-score matching was undergone following the nearest-neighbor matching method with a caliper size of 0.01 multiplied by the standard deviation for linearly transformed propensity scores. The balance of confounding variables in the matched groups was assessed by measuring their standardized mean differences between both groups. All standardized mean differences for the confounding variables were less than 0.05 (5%). Statistical analyses were performed using SAS statistical software (version 9.4; SAS Institute Inc. Cary, NC, USA). We considered a finding to be statistically significant if the two-sided *p*-value was less than 0.05.

## 3. Results

### 3.1. Patient Characteristics

Between 1 January 2011 and 31 December 2019, there were 45,506 patients newly diagnosed with HFS. Among these, through 1:4 matching using propensity scores by age, sex, socioeconomic status, residency, and comorbidities, 27,106 patients were finally enrolled in this study as the HFS group. A total of 108,424 participants were selected as the control group. Table 2 shows the characteristics of the HFS and control groups. No significant differences in variables were observed between the HFS and the control groups.

### 3.2. Mental Illness Risk in the HFS Group vs. the Control Group

The frequency of mental disorders among patients with HFS and the control group is shown in Table 3. Overall, patients with HFS were more likely to have a subsequent mental illness than the control group (8.5% and 6.5%, respectively, *p* < 0.001) within 90 days before and after diagnosis. Among mental illnesses, insomnia (46.2% vs. 13.0%, *p* < 0.001) was significantly more prevalent in the HFS group. Nevertheless, anxiety-related and stress-related disorder occurred similarly in both groups but was not statistically significant, and other mental illnesses were more prevalent in the control group. Eating disorder was too rare to be statistically significant. In the HFS group, insomnia was the most common disorder (46.2%), followed by anxiety-related and stress-related disorders (37.9%) and depressive disorder (14.4%). Anxiety-related and stress-related disorders (36.1%) and depressive disorders (32.3%) were frequently reported conditions in the control group, followed by insomnia (13.0%) and psychotic disorder (11.0%) (Figure 2). Figure 3 presents the odds ratio for mental illness according to the presence of HFS determined through logistic regression analysis. The patients with HFS had a significantly higher risk for mental illnesses than control patients (OR, 1.34; 95% CI, 1.27–1.41). Among the mental illnesses, insomnia was observed in 1064 patients of the HFS group, compared with 909 participants of the control group (OR, 4.55; 95% CI, 4.16–4.98), and compared to the control group. 

## 4. Discussion

Hemifacial spasm is a central nervous system disorder in which the muscles on one side of the face twitch involuntarily. As mentioned earlier, although HFS is not life-threatening, involuntary closure of the eyelids and twitching of the mouth can affect the quality of life of patients [9]. Some patients have even complained that their worst experience is social embarrassment due to facial disfigurement when interacting with strangers [4]. This problem may lead to low self-esteem and social isolation and, consequently, cause various mental illnesses. In a case-control study, Tan et al. reported that anxiety symptoms were significantly greater in patients with HFS than in healthy controls [6]. The authors also investigated the relationship between depression and HFS. In that cross-sectional study, the prevalence of depressive disorder in HFS patients was 16.7%, with younger women at higher risk. The severity of HFS was positively correlated with the severity of depressive symptoms [5]. In addition, Kim et al. reported that social phobia was common among patients with HFS, and microvascular decompression significantly improved facial disfigurement and social anxiety levels; these benefits were maintained for at least 36 months [7,8].

However, these studies were conducted with small sample sizes, and most of the patients participating in the study were assessed using a self-report questionnaire, such as the Hamilton Anxiety Rating Score, the Beck Depression Inventory, and the Liebowitz Social Anxiety Scale rather than a diagnosis by a psychiatrist according to DSM-5. Additionally, these patients tend to have more severe symptoms because they often visit the hospital for HFS treatment, such as microvascular decompression or botulinum toxin injection. Thus, these studies could not reflect the real-world prevalence of mental illnesses in HFS patients.

Hence, we investigated the association between HFS and mental illness through a nationwide population-based case-controlled study using the Korean NHIS database. To clarify the temporal causal relationship between HFS and mental illness, only psychiatric diagnosis codes registered within 90 days before and after the index date were defined as mental disorders. Usually, patients with HFS do not visit the hospital as soon as symptoms occur but are often diagnosed several months after symptoms appear. So, in the index date definition, we also included the onset of a mental disorder that occurred 90 days before the diagnosis of HFS. This definition has the disadvantage of not knowing the long-term effects of HFS on the incidence of mental illness. Nevertheless, this definition is essential to clarify the temporal causal relationship between HFS and mental illness. As mentioned above, HFS can act as a critical social stress factor, and it has a relatively immediate negative impact on the patient’s psychological and physical aspects.

In this study, we identified 2305 (8.5%) patients with mental disorders among 27,106 HFS patients, with insomnia being the most common among the mental illnesses, followed by anxiety-related and stress-related disorders within a relatively short period (Table 3). While designing the study, we expected depression or anxiety-related and stress-related disorders to be more dominant in HFS patients. However, contrary to our expectations, only insomnia was more frequent in patients diagnosed with HFS than in the control group.

Insomnia is a patient-reported problem characterized by difficulties initiating sleep or waking up from sleep during the night or earlier in the morning than one would like, with difficulty in resuming sleep. In a polysomnographic study, Inciril et al. reported that facial spasm continues during sleep in patients with HFS despite an apparent decrease, causing increased arousal and decreased sleep quality [10]. Additionally, insomnia also can be induced by a hyperarousal state experienced during the entire day [11]. Perlis et al. proposed the neurocognitive theory of insomnia [12]. This theory is based on the behavioral perspective that insomnia occurs acutely in association with predisposing and precipitating factors such as psychosocial stressors [13]. Although considered insignificant by some patients, as mentioned above, HFS can lead to catastrophic stressful conditions and a poor quality of life due to social embarrassment for the suffering individual. It means that patients with HFS are in a hyperarousal state due to worry about facial twitching, and this may lead to insomnia in the early stage. We, therefore, suggest that not only facial symptoms persist during sleep but hyperarousal states by facial spasms during the day also contribute to the development of insomnia.

The most noteworthy limitation of this study is that it could not guarantee the adequacy of the diagnosis and does not reflect the severity of the disease. We defined HFS and mental illness on the basis of ICD codes in insurance claims data. However, numerous studies using claims-based definitions have already been reported, and these administrative data have high specificity with variable sensitivity for diagnoses and medical conditions [14]. Because insurance data based on hospital visits were used, HFS and mental disorders may have been underestimated. Moreover, it is difficult to generalize because of the differences in the medical environment in Korea and other countries. Finally, the reason the prevalence of mental illnesses other than insomnia and anxiety-related and stress-related disorder was higher in the control group despite matching between the two groups requires further investigation. Despite these limitations, this study is meaningful as it is the first study to examine the relationship between mental illnesses, especially insomnia and HFS, using national claims-based data.

## 5. Conclusions

This study shows that patients diagnosed with HFS were significantly more likely to develop insomnia than a sociodemographic- and comorbidity-matched control group during a relatively short-term period.

## Figures and Tables

**Figure 1 jpm-13-00197-f001:**
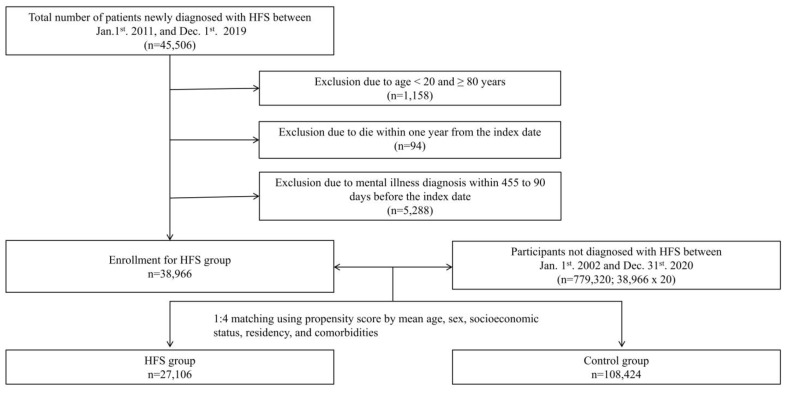
Schematic illustration of the participant selection from the Health Insurance Review and Assessment Service database of Korea.

**Figure 2 jpm-13-00197-f002:**
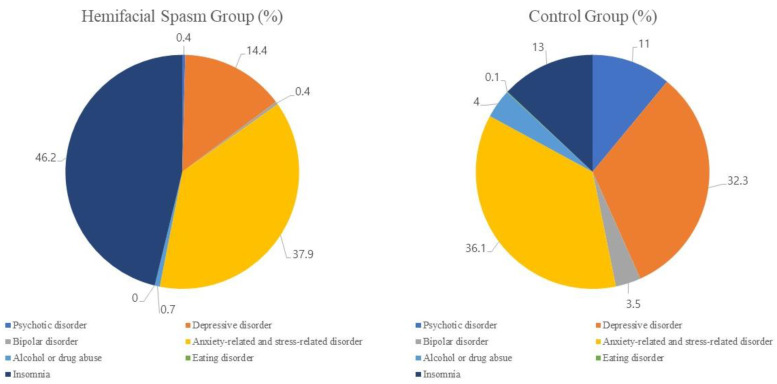
The distribution of mental illnesses in the hemifacial spasm group and the control group.

**Figure 3 jpm-13-00197-f003:**
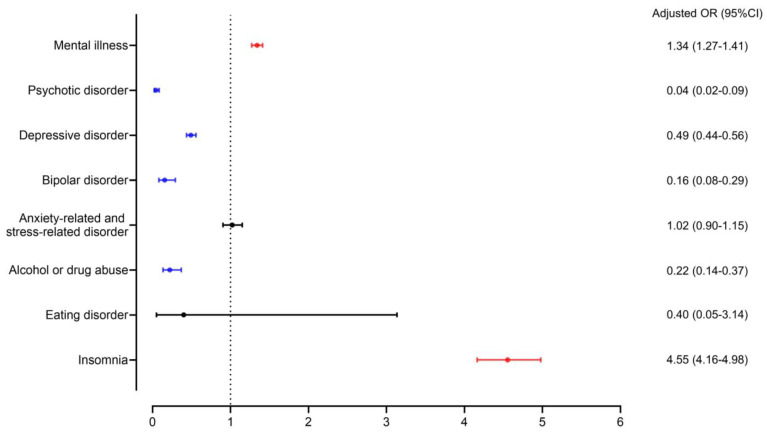
Propensity score-matched association of mental illnesses with hemifacial spasm.

**Table 1 jpm-13-00197-t001:** Working definitions derived from the insurance claims.

Psychiatric Disease	Working Definitions Based on ICD-10 *
Psychotic disorder	F20.X-F29.X
Depressive disorder	F32.X, F33.X, F34.X
Bipolar disorder	F30.X, F31.X
Anxiety-related and Stress-related disorder	F40.1, F41.X, F43.X
Alcohol or drug abuse	F10.X-F16.X, F18.X, F19.X
Eating disorder	F50.X
Insomnia	F51.0, G47.0

* International Classification of Diseases, 10th edition.

**Table 2 jpm-13-00197-t002:** Baseline characteristics of the study population.

Variable	Control Group	HFS Group	*p*-Value
	(n = 108,424)	(n = 27,106)	
Age (years)			0.222
Mean ± SD	55.72 ± 12.60	55.52 ± 12.70	
20–29	3876 (3.5%)	1010 (3.7%)	
30–39	8905 (8.2%)	2318 (8.6%)	
40–49	17,838 (16.5%)	4530 (16.7%)	
50–59	33,035 (30.5%)	8180 (30.2%)	
60–69	29,612 (27.3%)	7316 (27.0%)	
70–79	15,158 (14.0%)	3752 (13.8%)	
Sex			0.650
Male	37,449 (34.5%)	9402 (34.7%)	
Female	70,975 (65.5%)	17,704 (65.3%)	
Socioeconomic status			0.998
Fifth quintile (highest)	29,854 (27.5%)	7453 (27.5%)	
Fourth quintile	22,423 (20.7%)	5606 (20.7%)	
Third quintile	17,617 (16.3%)	4431 (16.3%)	
Second quintile	15,206 (14.0%)	3788 (14.0%)	
First quintile (lowest)	17,046 (15.7%)	4248 (15.7%)	
Covered by Medical aid	6278 (5.8%)	1580 (5.8%)	
Region of residence *			0.819
Seoul	24,420 (22.5%)	6147 (22.7%)	
Urban	30,412 (28.1%)	7563 (27.9%)	
Rural	53,592 (49.4%)	13,396 (49.4%)	
History of hypertension	40,742 (37.6%)	10,145 (37.4%)	0.650
History of diabetes mellitus	16,492 (15.2%)	4145 (15.3%)	0.739
History of dyslipidemia	38,874 (35.9%)	9765 (36.0%)	0.598
History of chronic renal disease	459 (0.4%)	181 (0.67%)	<0.001
Charlson Comorbidity Index			0.348
0	50,388 (46.5%)	12,553 (46.3%)	
1	31,294 (28.9%)	7755 (28.6%)	
≥2	26,742 (24.7%)	6798 (25.1%)	
Mental illnesses	7000 (6.5%)	2305 (8.5%)	<0.001

* Regions of residence were divided into 16 areas, according to the administrative district, before being reorganized into urban (Seoul, Busan, Daegu, Incheon, Gwangju, Daejeon, and Ulsan) and rural areas (Gyeonggi-do, Gangwon-do, Chungcheongbuk-do, Chungcheongnam-do, Jeollabuk-do, Jeollanam-do, Gyeongsangbuk-do, Gyeongsangnam-do, and Jeju).

**Table 3 jpm-13-00197-t003:** The frequency of mental illnesses among the patients with HFS and the control group.

Mental Illnesses	Control Group	HFS Group	*p*-Value
	(n = 70,000)	(n = 2305)	
Psychotic disorder	771 (11.0%)	9 (0.4%)	<0.001
Depressive disorder	2259 (32.3%)	332 (14.4%)	<0.001
Bipolar disorder	244 (3.5%)	10 (0.4%)	<0.001
Anxiety-related and Stress-related disorder	2527 (36.1%)	873 (37.9%)	0.682
Alcohol or drug abuse	280 (4.0%)	16 (0.7%)	<0.001
Eating disorder	10 (0.1%)	1 (0.0%)	0.366
Insomnia	909 (13.0%)	1064 (46.2%)	<0.001

## Data Availability

All of the raw and processed data were stored and can be accessed in our laboratory, which is supervised by the corresponding authors, D.S.C.

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
