# Peer review of "Early-Onset Insomnia among Patients with Hemifacial Spasm in South Korea: A Nationwide Cohort Study"

_jpm, 2023, doi:10.3390/jpm13020197_

Round 1

Reviewer 1 Report

The manuscript “Early-onset mental illnesses among patients with hemifacial spasm in South Korea: a nationwide cohort study” analyses the incidence of mental illnesses before and shortly after the diagnosis of HFS. It is an interesting study using a large sample, however the methodology used and the conclusions should be revised.

In first place, I don’t understand the reason why psychiatric diagnosis before the diagnosis of HFS are included. It seems reasonable, as it is commented in the discussion section, that to clarify the temporal causal relationship between HFS and mental illness only a short period of time after the diagnosis of HFS should be considered, however, including the 3 months prior to the diagnosis seems like it could confound the results.

In second place, it is also explained in the discussion that patients with HFS have continued facial spasms during sleep. The authors should state whether they consider it could be possible that the increase in insomnia incidence in this population might be due to a direct effect of such spasms in quality of sleep rather than the appearance of a co-occurring mental illness due to a hyperarousal state (a hypothesis that should be tested in another study…). Discussion section should be modified to address these issues.

Finally, conclusions are misleading since the results show only an increase in insomnia and not in any other mental illness in the sample, therefore stating that “that patients diagnosed with HFS were significantly more likely to develop mental illnesses such as insomnia and anxiety-related and stress-related disorders” is inadequate.

Author Response

Reviewer 1

Comments and Suggestions for Authors

The manuscript “Early-onset mental illnesses among patients with hemifacial spasm in South Korea: a nationwide cohort study” analyses the incidence of mental illnesses before and shortly after the diagnosis of HFS. It is an interesting study using a large sample, however the methodology used and the conclusions should be revised.

In first place, I don’t understand the reason why psychiatric diagnosis before the diagnosis of HFS are included. It seems reasonable, as it is commented in the discussion section, that to clarify the temporal causal relationship between HFS and mental illness only a short period of time after the diagnosis of HFS should be considered, however, including the 3 months prior to the diagnosis seems like it could confound the results.

: Thank you for your valuable comment. Usually, patients with HFS do not visit the hospital as soon as symptoms occur but are often diagnosed several months after symptoms appear. At first, they thought it was no big deal, but when the symptoms worsened, they felt abnormal and visited the hospital. Therefore, in the index date definition, we also included the onset of a mental disorder that occurred 90 days before the diagnosis of HFS. So we have addressed your comment by adding the following sentence to our discussion: “Usually, patients with HFS do not visit the hospital as soon as symptoms occur but are often diagnosed several months after symptoms appear. So, in the index date definition, we also included the onset of a mental disorder that occurred 90 days before the diagnosis of HFS.”

In second place, it is also explained in the discussion that patients with HFS have continued facial spasms during sleep. The authors should state whether they consider it could be possible that the increase in insomnia incidence in this population might be due to a direct effect of such spasms in quality of sleep rather than the appearance of a co-occurring mental illness due to a hyperarousal state (a hypothesis that should be tested in another study…). Discussion section should be modified to address these issues.

: We appreciate your advice and have corrected the discussion section as follows; ”Additionally, insomnia also can be induced by a hyperarousal state experienced during the entire day [11]. Perlis et al. proposed the neurocognitive theory of insomnia [12]. This theory is based on the behavioral perspective that insomnia occurs acutely in association with predisposing and precipitating factors such as psychosocial stressors [13]. Although considered insignificant by some patients, as mentioned above, HFS can lead to catastrophic stressful conditions and a poor quality of life due to social embarrassment for the suffering individual. It means that patients with HFS are in a hyperarousal state due to worry about facial twitching, and this may lead to insomnia in the early stage. We, therefore, suggest that not only facial symptoms persist during sleep, but also hyperarousal states by facial spasms during the day also contribute to the development of insomnia.

Finally, conclusions are misleading since the results show only an increase in insomnia and not in any other mental illness in the sample, therefore stating that “that patients diagnosed with HFS were significantly more likely to develop mental illnesses such as insomnia and anxiety-related and stress-related disorders” is inadequate.

: We have revised our title and conclusion section as follows; “Early-onset insomnia among patients with hemifacial spasm in South Korea: a nationwide cohort study”, “This study shows that patients diagnosed with HFS were significantly more likely to develop insomnia than a sociodemographic- and comorbidity-matched control group during a relatively short-term period

Reviewer 2 Report

The manuscript submitted by Kim et al. presented an interesting, data survey-based association study investigating the relationships between HFS and early-onset mental illnesses in South Korea. Generally, the manuscript is well-written, with solid data collection, analysis, and presentation. The outcome of the association study is also interesting, albeit not unexpected. Hence, I would recommend the acceptance and publication of the manuscript as a resource for the research community.

It may also be worth discussing the causative relationship between HFS and insomnia. Whilst previous research has revealed that both HFS and insomnia could be the result of genetic predispositions, whether the two traits have a fair share of the same genetic pool (say, the same set of genetic mutations lead to the clinical observations of HFS and insomnia), or one trait is the physiological result of the other (say, HFS causes involuntary facial muscle spasm and its subsequent disruption of sleep leads to reduced sleeping quality and insomnia) may worth a few words in discussion.

Author Response

Reviewer 2

Comments and Suggestions for Authors

The manuscript submitted by Kim et al. presented an interesting, data survey-based association study investigating the relationships between HFS and early-onset mental illnesses in South Korea. Generally, the manuscript is well-written, with solid data collection, analysis, and presentation. The outcome of the association study is also interesting, albeit not unexpected. Hence, I would recommend the acceptance and publication of the manuscript as a resource for the research community.

It may also be worth discussing the causative relationship between HFS and insomnia. Whilst previous research has revealed that both HFS and insomnia could be the result of genetic predispositions, whether the two traits have a fair share of the same genetic pool (say, the same set of genetic mutations lead to the clinical observations of HFS and insomnia), or one trait is the physiological result of the other (say, HFS causes involuntary facial muscle spasm and its subsequent disruption of sleep leads to reduced sleeping quality and insomnia) may worth a few words in discussion.

: Thank you for your decision. As mentioned above, we have corrected the discussion section as follows; ”Additionally, insomnia also can be induced by a hyperarousal state experienced during the entire day [11]. Perlis et al. proposed the neurocognitive theory of insomnia [12]. This theory is based on the behavioral perspective that insomnia occurs acutely in association with predisposing and precipitating factors such as psychosocial stressors [13]. Though considered to be benign by many people, as mentioned above, HFS can lead to catastrophic stressful conditions and a poor quality of life due to social embarrassment for the suffering individual. It means that patients with HFS are in a hyperarousal state due to worry about facial twitching, and this may lead to insomnia in the early stage. We, therefore, suggest that not only facial symptoms persist during sleep, but also hyperarousal states by facial spasms during the day also contribute to the development of insomnia.

Round 2

Reviewer 1 Report

All issues have been adressed